# The Magic of Optics—An Overview of Recent Advanced Terahertz Diffractive Optical Elements

**DOI:** 10.3390/s21010100

**Published:** 2020-12-26

**Authors:** Agnieszka Siemion

**Affiliations:** Faculty of Physics, Warsaw University of Technology, 75 Koszykowa, 00-662 Warsaw, Poland; agnieszka.siemion@pw.edu.pl

**Keywords:** diffractive optical elements, terahertz optics, beam shaping, 3D printing, convolution approach, on-axis and off-axis regime, neural-network-based optimization

## Abstract

Diffractive optical elements are well known for being not only flat but also lightweight, and are characterised by low attenuation. In different spectral ranges, they provide better efficiency than commonly used refractive lenses. An overview of the recently invented terahertz optical structures based on diffraction design is presented. The basic concepts of structure design together with various functioning of such elements are described. The methods for structure optimization are analysed and the new approach of using neural network is shown. The paper illustrates the variety of structures created by diffractive design and highlights optimization methods. Each structure has a particular complex transmittance that corresponds to the designed phase map. This precise control over the incident radiation phase changes is limited to the design wavelength. However, there are many ways to overcome this inconvenience allowing for broadband functioning.

## 1. Introduction

Diffractive optical elements (DOEs) are structures that introduce particular phase shifts of the incoming wave to create a desired phase profile of the outgoing beam. Such structures are defined by their transmittance which is defined by a complex function. This function is transformed into an optical element that diffracts the radiation to obtain a particular phase distribution after it. The complex transmittance describes how the amplitude and the phase of the incoming radiation will be changed after passing through the structure. Depending on the functioning of the element (focusing, redirecting, splitting, etc.) [1,2,3,4,5,6,7,8] such structures have different shapes. They can look like Fresnel lenses, spirals, maps of mountains, peacock eyes, hyperbolas or some other shapes, also looking like totally arbitrary random patterns [9,10,11,12]. This whole variety of shapes determines the way how the element will shape the incident radiation, focusing on one or more spots into a line segment, redirect, extend along the optical axis, form an arbitrary image and many more depending on the application needs.

DOEs can have different shapes, but the type of the structure matters as well. It determines how the desired phase distribution after the element is transformed to form a particular DOE transmittance. There are two main types of such coding—amplitude and phase. The former elements block part of the radiation and take advantage of diffraction at the edges of their not transparent fragments. The latter elements introduce particular phase shifts obtained by the propagation of the radiation in the material with larger refractive index (than surrounding). In this case, the thickness of such structures is strictly related to the refractive index of the used material and the design wavelength. Thus, the condition of introducing a particular phase delay by the structure results in the fact that DOEs are assumed as elements working in narrow-band regime.

Another possibility to precisely control the phase distribution is related to a slightly different type of optical elements. This group is based on the interaction of THz radiation with an array of resonators which form spatially dependent phase changes resulting in the required beam steering. Such elements are described as reflect-arrays, transmit-arrays, nonuniform metasurfaces or metamaterials. In recent years, the resonator-based optics is gaining the interest of researchers [13,14,15,16,17,18]. Although such devices also introduce a particular phase distribution corresponding to the phase delay map, they function according to different phenomena than described in this work diffractive elements. More information about different methods of terahertz beam forming was described in [19,20,21].

In many aspects, THz radiation range is peculiar due to the fact that it combines the two entangled worlds of optics and electronics. This is the region where solutions used in both these domains slightly stop to be effective. Antenna-based systems, multiplexers, waveguides, couplers, phased arrays, Luneburg lenses and other optical devices used for millimetre waves are meeting the world of mirrors, refractive and diffractive lenses, fibres, prisms, and gratings to form a unique mixture of optical and electronic phenomena. For different applications depending on the design frequency (lower or higher part of THz region), various optical solutions can be used. Similar to the use of emitters/sources and detectors with their roots in photonics or electronics, the optical devices are also transformed to THz region and are not favoring any design technique. Depending on the assumed use and the design frequency range, different solutions should be used, either approaching optical devices or taking advantage of electronic components. It should be underlined that diffraction optical elements are lightweight and thin. Additionally, they enable realization of the large aperture and small focal length elements. Moreover, they also are capable of strongly off-axis beam shaping and can form different shapes and can almost freely redirect the incident radiation. These features are quite attractive for further developments in THz imaging and spectroscopy.

## 2. Characteristic Features Concerning Manufacturing

Diffractive optical elements are used to shape the radiation in all spectral ranges; however each range has its peculiarities. They result from different material availability and their machining complexity level. Thus, in each range of radiation, different types of DOEs are common and mostly used. Even amplitude structures with the lowest possible efficiency, such as diffractive elements are commonly used in UV to focus the radiation [22]. For this frequency range there is the lack of materials introducing phase shifts that are transparent [23,24]. Thus, there is a need to optimize the thickness of diffractive elements (illustrated in Figure 1, the exact explanation of all mechanisms is given in the following sections of this review). Another aspect to consider is the resistance of materials to the high power of incident radiation [25].

In the case of visible light, the major drawback is that the human eye can see all lines corresponding to the phase discontinuities in the structure. The human eye is very sensitive to small changes of the refractive index. Therefore, DOEs are used e.g., in contact lenses (placed right after the pupil) rather than are used in corrective glasses [26,27,28,29]. In this spectral range, DOEs are also commonly used in cameras and imaging setups [30]. There is a whole variety of available optical materials, however manufacturing ensuring required resolution is costly (taking into account for example electron beam lithography, laser lithography, etc.).

Moving into longer wavelengths, elements designed for the infrared (IR) range of radiation suffer from the fact that only a few materials are mostly used in the IR optical industry. We can distinguish three groups depending on the refractive index value: around 1.5 (e.g., calcium fluoride, fused silica, magnesium fluoride, N-BK7 glass, potassium bromide, sodium chloride, and sapphire), more than 2 (e.g., zinc selenide and zinc sulfide) and more than 3 (e.g., germanium andsilicon). At the same time, it is worth noting that the refractive indices for visible materials vary between 1.45 and 2, while for IR they reach the value as high as 4. In such a case, substantial Fresnel losses take place and it means that around one-third of the incident radiation is reflected back from the surface of the element. Thus, there is a need to apply antireflection coatings [31,32] to reduce the amount of wasted reflected light.

The same situation occurs in the case of terahertz radiation high-refractive index materials such as e.g., germanium, silicon, gallium arsenide and sapphire. Fortunately, in THz range of radiation there are also a few materials with refractive index around 2 (e.g., quartz, fused silica). There is also a considerable group of materials with a refractive index of the value around 1.5 [33] such as various polymers, polyamides, paraffins, papers and many other unusual materials like for example natural stone [34], caramelized sucrose [35] or chocolate [36]. The most typical polymers are polyethylene (PE) and high density polyethylene (HDPE), Teflon (PTFE), polyolefine based on poly 4 methyl pentene-1 (TPX), Tsurupica. There is also a whole variety of 3D printed materials [37] like for example polyamide PA12 [6], high-impact polystyrene—HIPS [37], cyclic olefin copolymer—TOPAS [4,38] which are not only relatively transparent for terahertz radiation but also easy in manufacturing and thus commonly used to manufacture DOEs.

Moreover, in the case of THz range, the verification of the optical constants of the material is simply available thanks to Terahertz Time Domain Spectroscopy. This technique is a powerful and well developed one and allows extracting different information about the tested material such as refractive index and absorption coefficient [39,40,41]. These two parameters are crucial in designing diffractive optical elements to determine its attenuation and proper structure thickness [42] to introduce a particular phase delay:(1)$$\varphi_{DOE}\left(x,y\right)=-\frac{2\pi }{\lambda_{DWL}}\left(n-n_{0}\right)h_{DOE}\left(x,y\right),$$
where $\varphi_{DOE}\left(x,y\right)$ is the phase distribution, $h_{DOE}\left(x,y\right)$ is the thickness of the structure in each point of the structure, $\lambda_{DWL}$ is the design wavelength ($k=\frac{2\pi }{\lambda_{DWL}}$ is the wave number), *n* and $n_{0}$ are refractive indices of the structure and surrounding (usually air) materials. Such a phase distribution is later coded into the transmittance of the DOE and the value of both refractive index and absorption coefficient must be known and taken into account to manufacture the designed structure properly to assure the highest possible efficiency. The optical constants of exemplary materials available for extrusion-based 3D printing techniques [43,44,45] are given in Figure 2. It can be clearly seen that the most commonly used polylactide (PLA) is characterised by a much higher absorption coefficient than for example PA12. The higher the frequency, the larger attenuation coefficient can be treated as a rule for THz optical materials (maybe except germanium, which has a local minimum in the attenuation curve around 1.3 THz [33]).

The incredible availability of different materials and their easiness in manufacturing had a tremendous influence on the recent development of THz diffractive optical elements. Even in the case of reflective optics, 3D printing methods [46] are used and assure a good quality. The phenomenon of THz optical elements, especially for lower frequency range, is related to relatively long wavelengths in the range of single millimeters or tenths of a millimeter. Assuming that very good quality optics has the roughness corresponding to λ/20 or even λ/10 in case of lower frequency THz range, such a precision of manufacturing is easily fulfilled by 3D printing methods even those simplest based on extrusion.

Unfortunately, for higher frequencies in THz range, the 3D printed materials start to attenuate too much, which prevents their application. It should also be highlighted that the thickness of the diffractive structure in such a case is thinner as it is related to wavelength. Thus, even a high attenuation coefficient value does not have to make a very significant contribution to the overall attenuation introduced by the structure which is resulting from Beer-Lambert’s law. The major drawback in the higher THz frequency range results from the limited resolution of 3D printing techniques, which starts to be insufficient here. Moreover, the aspect ratio (height over width) of printed details is too big for proper manufacturing. Struggling with technology to obtain a higher resolution leads to more cost demanding solutions like etching [47] or ablation [48,49]. However, the material also changes—mostly for silicon. As was already been mentioned, there is the issue of large Fresnel losses connected with high refractive index of silicon and consequently the antireflection structures are required [50]. It should be underlined that a particular pyramid -like surface can be formed in the process of laser ablation and this surface acts as an antireflection structure [51].

The other solution in the case of DOEs working at higher THz frequencies is the use of high order kinoform [52,53,54] as a different way of phase map coding. Such structures introduce the maximal phase shift equal not to 2π, but to its multiplicity. They are thicker, but what is crucial, the size of the zones or structure details increases proportionally to the order of kinoform. Therefore, the resolution of 3D printing meets the manufacturing criteria. The resolution can be assumed to be mostly 0.05 mm even though it is different depending on the particular technique and the device. To reach the goal of obtaining high DOEs efficiency the low absorbing material is needed, which can be 3D-printed or melted and poured into a 3D printed form. In Figure 3 the optical constants of different THz alternative materials are illustrated. Good properties of some kinds of paraffin in comparison with other materials can also be observed. HDPE was also plotted as the reference material. Many different 3D printed materials were analyzed due to the simplicity and accessibility of manufacturing DOEs for THz range. Moreover, the direct 3D printing of all metal terahertz components has also been demonstrated in the literature [55,56].

In the context of the described issues, it is important that in THz range there is a whole variety of different materials suitable to manufacture optics. Due to their low absorption coefficient values in many cases there is no need to assume attenuation. However, taking into account materials with higher attenuation, there may be the need to optimize the thickness of the DOE not only considering the relation to the refractive index value and design wavelength but also the attenuation. Such a procedure is used in the case of UV optics when the structure thickness is thinner than the value calculated from the diffractive optics assumption [23]. It introduces a smaller phase shift but also according to Beer-Lambert’s law—attenuates less which can be also used for THz range [56].

Finally, it is worth noticing that all these considerations about diffractive optics and materials are related to fitting the designing procedure to a particular frequency range and its characteristic features. THz range of radiation delivers immense possibilities for designing and manufacturing diffractive (and not only) optical elements. Thanks to the diversity of available techniques, many advanced optical elements could be realized and experimentally tested.

## 3. Classic DOE Design

In general, the design of diffractive optical elements comes down to creating a structure with such a transmittance $T_{DOE}\left(x,y\right)$ which will form a particular phase distribution φ(x,y) of radiation after passing this element. φ(x,y) can be binarized and form amplitude element $A_{DOE}\left(x,y\right)$ with transparent and radiation blocking areas or form a phase element $\varphi_{DOE}\left(x,y\right)$ consisting of areas introducing only two different phase shifts. Thus, the desired phase distribution is coded in the form of the required transmittance of DOE—$T_{DOE}\left(x,y\right)=A_{DOE}\left(x,y\right)e^{-i\varphi_{DOE}\left(x,y\right)}$ [57]. Of course φ(x,y) does not have to be binarized, it could be coded into transmittance with different shapes, such as sine-like, step-like or saw-blade (blazed, corresponding to kinoform) [21,58,59].

Depending on this type of coding of φ(x,y) the designed element will have different diffraction efficiency η, regardless of attenuation. The type of coding defines the maximal diffraction efficiency of DOE assuming proper element thickness. The arising question concerns the origin of values describing η for different coding types. The dependence is straightforward—the diffraction efficiency of the DOE is equal to the diffraction efficiency of the first (m=1) order of the diffraction grating corresponding to the coding type. Therefore, for example, binary amplitude coding will have the maximal η equal to 10.1% which is the $\eta_{m=1}$ of the amplitude binary diffraction grating. All maximal η values for each coding type are given in Table 1 and in [21,30,60,61,62].

The examples of different grating types are illustrated schematically in Figure 4a (insets on the left panel). Each grating redirects different amounts of energy in different orders of diffraction (here, only ±2nd, ±1st and 0th are marked). For some gratings, not all orders are formed, and some of the orders are symmetric while the others are not. The rule defining how much of the incident radiation will be redirected in the particular m−th order of diffraction is defined by [63]:(2)$$\eta_{m}=\frac{I_{m}}{I_{all}}=\frac{I_{all}{\left|G_{m}\right|}^{2}}{I_{all}}={\left|G_{m}\right|}^{2},$$
where $I_{all}$ is the intensity of radiation illuminating the grating (total incident radiation), $I_{m}$ is the intensity of radiation redirected into m−th order of diffraction and $G_{m}$ is the amplitude coefficient of the expansion of the function *f*(*x*) describing the periodic grating with the period d into the Fourier series given as [42]:(3)$$\eta_{m}={\left|G_{m}\right|}^{2}=|\frac{1}{d}\int_{-\frac{d}{2}}^{\frac{d}{2}}f\left(x\right)e^{-2\pi i\frac{mx}{d}}dx{|}^{2},$$
(4)$$\eta_{0}={\left|G_{0}\right|}^{2}=|\frac{1}{d}\int_{-\frac{d}{2}}^{\frac{d}{2}}f(x)dx{|}^{2}.$$

Another striking issue is related to the question of what happens with the rest of the radiation? In the case of diffraction grating, it is the radiation that is redirected into different orders, which is illustrated in Figure 4a. As it can be seen, 0th remains unbent (corresponding to violet color), positive orders are bent in one direction at different angles (light violet) and negative orders are bent in another direction, also at different angles (dark violet). The connection between the diffraction efficiency of gratings and DOEs is illustrated schematically in Figure 4b. Assuming that our DOE is a simple focusing lens, the part of the incident radiation (ideal plane wave) will be focused at a distance equal to the focal length—*f*. The amount of intensity focused by DOE lens corresponds to the amount of intensity redirected into the first order of diffraction of the grating (marked with light violet color and +1st ). The radiation related to all subsequent orders is focused at distances equal to $\frac{f}{m}$ (marked with light violet). All negative orders of diffraction of the grating describe the radiation that is diverged by the DOE lens and are described with virtual focal spots formed at distances $\frac{f}{m}$ before the structure (marked with dark violet, and the virtual rays with dashed lines). The radiation that is called 0th order of diffraction is the radiation that passes through the structure without any influence from the structure (marked with violet). It is clearly seen that only a part of the radiation is focused at a proper distance equal to *f*. All the rest is either focused closer to the structure (light violet), or diverged (dark violet) or remains unchanged after passing the structure (violet). Thus, all mentioned orders of grating or DOE are the unwanted noise.

If the manufacturing technique and available materials allow creating a kinoform lens, then it is the best solution. Such a solution will assure the highest possible and equal to 100% diffraction efficiency. All other coding types will suffer from similar attenuation of the structure but will be characterized by much smaller diffraction efficiencies. Only multistep phase structures can approach the theoretical diffraction efficiency of almost 100%.

## 4. Features of THz Design

Terahertz range of radiation is very particular as far as many issues are concerned. As it is the link between the world of optics and electronics, it benefits from these two fields. However, some physical phenomena become also more crucial for THz range of radiation than for other spectral ranges.

Focusing on diffractive optics, the THz range gives a large opportunity of easy manufacturing of different optical elements. In many cases, simple 3D printing and milling (supported by Computerized Numerical Control), laser cutting, or much more complicated ablation or etching can be used. Thanks to all these manufacturing opportunities, it is highly advisable to use coding with a continuous phase profile—in the form of so-called kinoform. From the theoretical point of view, kinoforms have 100% of diffraction efficiency which assures that the incident radiation is not wasted as a result of the coding method. In such a case, we must consider Fresnel losses and attenuation. The former is strictly related to the refractive index difference on the interface between two media and is given by the equation describing the reflection loss at normal incidence:(5)$$r_{\lambda }=\frac{{\left(n_{\lambda }-1\right)}^{2}}{{\left(n_{\lambda }+1\right)}^{2}},$$
where $n_{\lambda }$ is the ratio of two refractive indices between two media. This expression assumes not polarised radiation and includes the average of the reflection coefficients for both s- and p-polarisation states of the incident radiation. The larger the incidence angle is, the stronger effect of Fresnel losses is observed. Moreover, two additional effects concerning the angle of incidence of the illuminating beam must be considered. These effects are described by the cosine loss mechanism [64] (illumination at an angle different than normal results in a cosinusoidal decrease of the radiation flux) and the shadow effect [4,65].

The attenuation introduced by the structure depends on the attenuation coefficient and thickness of the structure. According to the Beer-Lambert’s law [33] a transmitted part of the incident monochromatic wave (not taking into account reflection losses) can be defined as:(6)$$E_{trans}=E_{inc}e^{inkd}e^{-k\kappa d},$$
where $E_{inc}$ is the incident beam, *n* is refractive index, *k* is the wave number, *d* is the thickness of the material and κ is the extinction coefficient. κ is related to the attenuation coefficient with the following equation:(7)$$\alpha =\frac{2\omega \kappa }{c}=\frac{4\pi \kappa }{\lambda_{0}},$$
where $\lambda_{0}$ is the wavelength in vacuum. The first exponential function describes the phase retardation introduced by the structure, while the second one describes the attenuation introduced by the structure (not taking into account reflection losses).

Designing THz optical elements (not only diffractive) encounters two major problems: reflection losses and attenuation of the material. These two problems influence the effective efficiency significantly. Nevertheless, in comparison to all other ranges of electromagnetic radiation—terahertz is characterized by simple and cost-effective manufacturing of optical elements. That is why it provides infinite opportunities of possible applications. However, there are also some issues related to this range of radiation such as the relation between the wavelength and the aperture size of the optical element. It should be underlined that for example for visible optics a 50-mm aperture system corresponds to around 100,000 wavelengths (assuming λ= 0.5 m), while for THz it is only 100 (for λ= 0.5 mm). The arising question is about the importance and actual meaning of this fact. The explanation is straight forward and is related to the diffraction phenomenon and so-called diffraction zones. The propagation of radiation is governed by the fact that the incident wave is being diffracted at obstacles and these secondary waves interfere with each other forming different patterns. The shape and characteristic features of such patterns are related to the size of the obstacle (aperture)—*D* and the observation distance—*z*. These two values together with the wavelength λ allow defining the following diffraction zones:Zone very close to the aperture—z≈λ and propagation is described by Rayleigh-Sommerfeld integral;Non-paraxial zone—z>>λ and z<D and propagation is described by Sommerfeld integral;Paraxial Fresnel zone (near-field)—$z\in (D,\frac{D^{2}}{4\lambda })$ where the paraxial Fresnel integral is used;Fraunhofer zone (far-field)—$z>\frac{D^{2}}{4\lambda }$ and propagation is described by Fraunhofer integral.

The border between the zones can also be described by Fresnel number $F=\frac{A}{\lambda z}$, where *A* is the area of the aperture. Of course, the zones merge smoothly into one another and there is no sharp border, but we can distinguish two regions with different behaviour—near-field and far-field. The former is related to changes in the diffraction pattern as bright and dark areas are changing during propagation. Here, we can observe Fresnel diffraction patterns and self-imaging phenomenon also called the Talbot effect. In the far-field zone, the intensity is described as the Fourier transform of the aperture function with the relevant scaling factor. This factor is also called a convolution kernel consisting of a constant exponential factor dependent on *z* and a quadric phase exponential factor.

Terahertz range of radiation is characterized by relatively long wavelengths, so using standard size optics is related to significant diffraction effects related mostly to near-field diffraction. To illustrate this problem, a simulation of imaging in 4f optical setup was conducted for 0.3 THz in case of 50-mm-diameter (and 75-mm-diameter) optical elements (Figure 5d,c) and assuming no additional element aperture (only size of the calculation matrix—Figure 5b). A 50-mm aperture (and 75-mm diameter) size is marked by the red circular dashed line. In Figure 5c,d the concentric rings typical for near-field diffraction are clearly visible, while in Figure 5b only a diffraction pattern resulting from the rectangular calculation matrix is visible. The fact that optical elements have limited apertures is the main reason of using point-to-point imaging, which means that the radiation is focused on the sample and then focused again on the detector. Such a setup allows creating the image only by shifting the sample but gives the opportunity to suppress negative diffraction effects. The necessity of forming uniform illumination was noticed [66] which resulted in forming top-hat beam with two refractive or two diffractive structures. The experimental evaluation clearly indicated a better and more uniform beam while using diffractive elements. Nevertheless, in real applications and instantaneous THz scanning, a plane-to-plane imaging must be carried out and thus a large aperture optical elements are crucial.

Hence, to develop and enable real applications of THz optical systems, there is a great demand for large aperture optics—which in the case of refractive design becomes extremely thick and in the case of off-axis elements the situation worsens. That is why the diffractive optical elements are becoming a serious player in this field due to their reduced thickness and increased transparency. Moreover, large aperture systems are easily available, especially due to the possibility of using 3D printing techniques with relevant materials. However, for higher THz frequencies, such manufacturing methods start to suffer from too low resolution, so the use of high order kinoforms is advisable. At the same time, kinoforms similarly to multilevel diffractive lenses (MDLs) function for broadband radiation. MDLs will be described later in this review.

In the next sections, the different THz DOE designs will be presented. The current trends in THz optics including recent (last 5 years—2016–2020) research in the field are taken into account.

## 5. Efficient Focusing of THz Radiation

Converging terahertz diffractive lenses were designed already in the 1960s and have been continuously developed since that time. Presently, simple Fresnel zone plates or Fresnel lenses are still in the area of research interest even as binary amplitude structures (called Soret Fresnel zone plates) [49,67]. They allow obtaining elements with large numerical aperture values (NA) and assure a compact optical setup configuration in comparison to refractive or reflective counterparts. Here, direct laser writing [68] (related to ablating the unwanted parts of the substrate layer) ensures proper manufacturing resolution. However, such a technique can be also used to manufacture phase structures with different number of phase steps [3,69,70] or even kinoform structures [71]. Laser ablation is mostly related to using high-resistance silicon plates or metals. Using silicon has one distinct disadvantage which is large Fresnel losses resulting from high refractive index value (n=3.4 in all THz range). However, this negative effect can be reduced significantly by using antireflection structures on the surfaces of manufactured elements [51]. Moreover, silicon is the most transparent material in the THz range and additionally it has the smallest dispersion (the refractive index value is almost not dependent on the frequency in this range—it has the same value up to 3rd place after the comma).

Lately, not only laser ablation has been used to manufacture diffractive lenses for THz range. Using a three-axis milling technique on slabs of a polymer (Zeonex) which has one of the smallest attenuation [72] is also the interesting solution. Here, milling has been applied to manufacture binary and four-phase-step lenses (as single- or double-sided structures).

Due to their characteristic features, THz diffractive optical elements can be manufactured using a whole variety of different techniques which have already been mentioned before. Some examples of focusing and redirecting optical structures together with their description are illustrated in Figure 6.

One of the greatest challenges in optics is to obtain a small focal spot to concentrate energy efficiently or to obtain good resolution in point-to-point imaging [73]. The focal spot size cannot be decreased to be an ideal point—such a situation would be possible in the case of having infinite size optical elements and ideal plane waves. In reality, the size of the focal spot is diffraction limited or stays within the Abbe limit for paraxial and non-paraxial approaches, respectively. The former is determined by the Airy disc size *r* described by the equation:(8)$$r=1.22\lambda \frac{f}{d},$$
where λ is the wavelength of radiation, *f* is the focal length of the structure and *d* is its diameter. The Abbe limit is defined for non-paraxial approach and depends on the wavelength of the radiation λ and the numerical aperture NA which is given by the equation:(9)$$r=\frac{\lambda }{2NA}=\frac{\lambda }{2nsin\theta },$$
where *n* is the refractive index of the medium and θ is the half angle of convergence of the focused wave.

Reducing the focal spot size can be realized by decreasing the focal length to diameter ratio (called F-number). However, while approaching a value equal to 1 it is crucial to assume non-paraxial approach. Moreover, by additional phase corrections resulting from the fact that the structure has a physical thickness (it is not an ideal infinitely thin element) the signal to noise ratio in the focal plane can be significantly augmented [74]. Diffractive optics gives the remarkable opportunity to increase the numerical aperture (NA) value of the structure (decrease F-number) to values that are not achievable in refractive and reflective approaches remaining compact element architecture [70]. Making the focal length shorter and element diameter larger leads to extreme off-axis functioning. Thus, the significant shadow effect appears for the outer zones resulting in a tremendous decrease of the element’s focusing efficiency. Radiation incident at a large angle is totally internally reflected inside the outer zones and instead of being focused, is diverged. Such a phenomenon can be prevented by, for example, using the inverse design of outer zones [2].

Not only a typical diffractive design is of interest to researchers. The structure composed of various holes located on every other structure zone, called a THz sieve, can successfully focus the radiation both in amplitude and phase coding [75]. Moreover, good focusing properties were obtained by using a subwavelength concentric ring structure array. The focal spot was smaller than for the reference refractive lens, but the side lobes were slightly larger. Nevertheless, alternative design approaches are more and more common to reduce a focal spot size.

Sub-diffraction focusing was also achieved by two types of effective super-oscillatory lenses [76,77,78]. Photo-lithography was used to manufacture two structures: one being arrays of metaatoms and another consisting of many concentric rings. In both cases, a focal spot smaller than the diffraction limit was obtained (for 0.1 THz) thanks to the use of super-oscillatory lenses with the smallest slit width of $\frac{\lambda }{6}$.

Focusing of the radiation does not have to be limited to a single focal spot but can also be understood as forming multiple focal spots (one after another), creating focal curves of different shapes or functioning for a broader range of frequencies. The Fibonacci lens design [79,80] was used to form bifocal diffractive lenses realized as binary structures with zone widths and position calculated according to the golden ratio [81]. Such elements were used to point-to-point imaging of volume objects—two planes corresponding to two focal lengths were imaged simultaneously forming an image with wavelength resolution. The next group describes optical elements forming focal curves of different shapes. The diffractive optical element design is closely related to Fourier optics, which makes the connection between the DOE plane and focal plane described by the Fourier transform with some scaling factor. Many analytical relations can be described [82] and focal curves—lines with different lengths and orientations, circles, hearts or other shapes—can be formed [9,42]. In the case when there is no analytical equation relating the focal and structure planes, a synthetic Fresnel hologram can be calculated forming any arbitrary shape. Moreover, the image can be formed in two or more planes behind the structure and can be larger than the aperture of the structure [10], which is a tremendous achievement. Shaping the incident radiation into different curves can improve THz signal acquisition and collection. It is of key importance in the fields of scanning, tomography, beam redirecting, coupling into detector and many others including using diffractive elements in matrices suited to a particular detector (or emitter) array [83].

It is an interesting idea to use diffractive structures with extended depth of focus as optical elements focusing at a particular distance the radiation with a broader spectral range and maintaining small thickness [84,85]. However, in recent years, a vast majority of optical elements working broadband have been designed using metamaterials [76,86,87].

Moreover, simple DOEs can be used to control phase delay and shape waveforms in the THz range [88] realized by wedge-like or spiral-like structures. The latter structures are also used to form non-diffractive beams described in the next section.

## 6. Non-Diffractive Beams

Simple 3D printing manufacturing methods and prism-like refractive structures can be used to form diffraction-free structured beams [89]. However, such structures are bulky and attenuate much more than diffractive optical elements. In the field of creating non-diffractive beams, there is a continuous development and novel ideas are dynamically invented. Regardless of the exact transmittance of the structure or structures generating non-diffractive beams, the following types of beams can be distinguished: Airy, Bessel and optical vortex, but also others such as Mathieu [90] or caleidoscopic ones [91].

The intensity patterns observed along the optical axis corresponding to Airy, Bessel and optical vortex beams are illustrated in Figure 7. Of course, their particular shapes depend on the simulation parameters and constants defined to calculate the phase distributions of the structures generating particular beams. The Bessel and vortex beams propagate symmetrically along the optical axis (Figure 7b,c—upper panel). While Airy beam creates bow-like intensity pattern that converges to propagate along the optical axis after some distance (Figure 7a—upper panel). The lower panel illustrates the cross-sections of the intensity across xy planes marked with red dashed lines on the yz intensity patterns (illustrated on the upper panel).

The Airy beam can be formed with two diffractive elements tear-like structure and a converging lens. The former is introducing the phase delays described by equation [92,93]:(10)$$\varphi_{Airy}\left(x,y\right)=a_{1}\left(x^{3}+y^{3}\right)-a_{2}\frac{\pi }{\lambda f}\left(x^{2}+y^{2}\right),$$
where λ is the wavelength, *f* is the distance between structures, *x* and *y* are coordinates in the structure plane, $a_{1}$ and $a_{2}$ are scaling constants.

Bessel beams can be generated by structures (refractive or diffractive) called axicons, which in the refractive version looks like a cone and its diffractive counterpart is a circular equidistant grating that can be described with an introduced phase retardation:(11)$$\varphi_{axi}\left(r\right)=-ksin\left(\theta \right)r,$$
where $k=\frac{2\pi }{\lambda }$ is the wave number, *r* is radial coordinate in structure plane and θ is the half-cone angle (the inclination angle with respect to the optical axis of the wavefront transformed by the axicon). Of course, there are also more complex equations describing axicon structures depending on the intensity distribution. Their diffractive design to form Bessel beams was proposed using 4-level structure [94] or kinoform [95]. The comparison between refractive and diffractive approaches is described in [95].

Another approach is related to designing spiral-like lens to form Bessel beams (also of higher order) that carry the orbital angular momentum (OAM) [96]. For high power radiation (for example coming from Free-Electron-Laser, FEL) the beam can be shaped using silicon phase spiral masks [97,98]. The analysis of non-diffractive Bessel vortex beams with different topological charges was conducted [98] for the structure introducing phase retardation:(12)$$\varphi_{vortex}\left(r\right)=-kr_{d}\left(\frac{l\Phi }{2\pi }+\frac{\beta }{2}\right),$$
where $r_{d}$ is the axicon period also related to the angle of diffraction of the incident beam and Φ corresponds to the azimuthal angle in the structure’s plane. The parameter *l* denotes the number of branches in the spiral structure with the sign determining the direction of the spiral rotation, while β=0,1,⋯,2l−1.

Vortex beams can also be created by helical-based elements combined with converging lens distributions with different angular area division [99,100]. The radius and width of the vortex allow controling the topological charge which results in efficient forming of multiplexed THz vortex rings. Using the described structures introduces the unchanged ring radius in the vortex beam for a long propagation distance. Even more advanced diffractive structures can be used to discriminate OAM modes in vortex beams, which could be used to multiplex or demultiplex different OAM channels efficiently [101]. The immense interest in vortex beams generation is assigned to their potential application in wireless communications. Many OAM modes can be coupled into a fiber and thus increase the capacity of THz communication systems significantly [102].

## 7. Redirecting of Incident Radiation

The diffraction grating is the first association coming to mind when thinking about the redirecting of radiation. For THz range, they can be easily manufactured by 3D printing and after metalization they can work in a reflection mode [7] or if a proper material is used, they can work in a transmission mode [4]. As mentioned before such a manufacturing technique has a sufficient resolution to produce blazed gratings. Even using a low-cost base material for THz optics such as paper tissue was presented on the example of a blazed grating [8].

All diffraction gratings must introduce a particular phase shift to maximize the possible diffraction efficiency. In the case of high-attenuating materials, the total efficiency can be larger for the grating with smaller phase shift but also smaller thickness. Both these values can be optimized to provide as large efficiency as possible for a particular grating type and used material [23,56]. Changing grating height was also used in the reflection grating to modify the ratio of intensities redirected between 0th and ±1st orders [103] by a set of gratings with different step heights. Thus, such a predefined beam steering was used in real-time multichannel Fourier-transform spectroscopy [103]. Metal grating was also used to shift the deflection angle of an incident beam thanks to the variable-period diffraction grating for terahertz frequencies made from a 3D printed spring-like holder and metal stripes [104].

A 2-dimensional grating with trapezoidal elements was designed for instantaneous spectral measurements [105]. The idea used also in X-ray range of radiation assumes single-order diffraction functioning in the ultrabroadband terahertz region. Such a grating has dispersive properties for different frequencies in the broadband incident spectrum, which results in the separation of particular spectral components.

It should be underlined that redirecting radiation can be realized as splitting different frequencies from the incident broadband beam, but also redirecting one single frequency beam into a matrix of points. The latter can be realized by optimizing the diffraction grating design to form a particular number of diffraction orders [5,106] or implementing structures called Dammann gratings [5,107,108]. The exemplary structures redirecting and splitting radiation are illustrated in Figure 8. They are based on lenses and different types of gratings. Such matrices of focal points can be used to optically pump an array of detectors (like for example superconducting hot electron beam mixers commonly used in astronomic THz observations). Moreover, bearing in mind that splitting one incident beam into many focal points is possible, so similarly a matrix of sources can be joined into one output beam which will increase its total power. Concluding, such fan-out elements can be used to join a matrix of relatively weak sources into one strong beam.

Another issue concerning the redirection of the radiation is related to bending the incident beam and focusing it at some angle according to the optical axis of the designed structure. Using effective grating theory gives very promising results to design such all-dielectric binary off-axis diffractive lens. The local period of the grating is described by the elliptical set of grooves over the area of the structure [109]. The incidence angle was assumed to be different than normal, which resulted in a more complex configuration. The structure with similar idea of functioning can be designed and described by iterative algorithms based on the convolution approach of light propagation. Iterations are necessary to increase the efficiency of the designed structure that can be understood as a kind of synthetic hologram (here using an iterative algorithm resembling ping-pong or Iterative Fourier Transform Algorithm—IFTA—approaches) [6]. The depicted geometry and design can be used for minimizing the in-reflection imaging setups thanks to the remarkable possibility of increasing NA of the system by diffractive design approach.

## 8. Optimization Algorithms and Neural Network Approach

Competing for better efficiency, resolution, broadband working or tailoring to a particular application leads to developing and optimizing the designed optical algorithms. The off-axis redirecting and focusing of THz radiation can be corrected by iterative redesigning of the diffractive structure [6] (described in the previous section). Moreover, the optimization can be conducted based on modified search algorithms. The gradient descent optimization based on a modified binary search algorithm is used to define the shape of multilevel phase structures focusing the radiation into point or line [110,111,112,113]. It allows creating 1D and 2D converging lenses with large NA values for both narrow and broadband focusing, which are error-tolerant and efficient. Moreover, spectral splitting at different spatial points can be optimized in a way which can find interesting applications in spectrometers. Of course, the optimization of phase transmittance of the diffractive element does not have to be limited to only simple optical devices such as lenses but can be also performed for such complex structures as holograms [113]. Moreover, in the case of holograms an iterative method of optimization can give also very good results [10].

Neural network architecture was a remarkable revolution in the context of computer algorithms. It was the starting point in the path where computers began to be taught by examples and generalize this knowledge. Such a new approach resulted in creating solutions not possible to obtain by conventional algorithms or equations. Performing complex decisions based on drawing conclusions was a characteristic feature of human beings but owing to the neural network framework it became an attribute acquired by a computer. In terahertz optical applications, a neural network design can be implemented in a variety of possibilities. However, so far it has been used for example to classify elements or to optimize distributions defined and tailored for particular needs.

The first considered here neural network application in the THz range described machine learning by demonstrating an all-optical diffractive deep neural network. The idea was to discriminate the input image by an optical system which consisted of a set of passive phase distributions calculated by neural network [114]. The 3D-printed structures created a diffractive neural network that could not only classify different images but also perform the function of an imaging lens in a terahertz spectrum. Such an approach can significantly increase the usability of optical systems in image analysis, detection of particular features, classification of analyzed items and performing specific tasks dedicated to neural network approach.

A similar scheme of diffractive optical network realized by 3D-printed structures was used to process simultaneously the broadband signal generated by THz TDS (Time Domain Spectroscopy) system. The predefined tasks could be realized using only optical elements that were designed using deep learning. Different multilayer diffractive optical systems were designed to perform the operations corresponding to various spectral filters or spatially controlled wavelength de-multiplexing [115]. It should be underlined that the use of neural-network-based learning algorithms can be expanded to multiwavelength operation and taking advantage of processing separate frequencies from broadband sources. Such a filtering of different frequencies can be applied in multispectral imaging, beam filtering and redirecting, multiple-input-multiple-output systems and many more.

More and more advanced optical structures require not only well-suited designs but also additional optimization techniques to enable desired functioning. Holograms are one of the most advanced optical structures which are mostly designed by back propagation methods, iterative design (like Gerchberg-Saxton or ping-pong algorithms) [10], but also neural network approach [116]. This last method uses a predefined target pattern and initial light field distribution to train a diffractive neural network that will enable to create a quantified phase profile. The designed and manufactured by 3D printing structure formed an on-axis image at a particular distance.

However, the quest for efficient off-axis functioning optical devices is especially important in THz range. A side by side comparison is useful to illustrate the issues that must be overcome in the non-paraxial design. The simplest hologram is a hologram of a point—which corresponds to the focusing lens. All other objects can be created from single points. It makes it easy to understand and realize hologram distribution calculations as the integration of signals originating from each object point separately. In case of the large hologram size and the small distance of image formation, an off-axis design is required. Thus, the simple back propagation method starts to be inefficient which is seen in Figure 9a. The calculated phase distribution of the diffractive structure is shown as a grey scale image (lower panel) and the corresponding reconstructed intensity pattern is shown in the color palette (upper panel) corresponding to the area of the structure marked with a red rectangle. The input object consisted of two Gaussian-shaped spots with the total diameter of 3 mm and being shifted from the main optical axis 25 mm and 50 mm, respectively. They were formed at a distance of 100 mm behind the structure, which corresponded to deflection angles equal to $14^{\circ }$ and $27^{\circ }$ respectively. The whole distribution corresponds to the size of 64 mm. As is illustrated in Figure 9a, the not diffracted component is forming the unwanted focal spot called DC term or 0th order of diffraction. Therefore, an iterative design can be used based on Gerchberg-Saxton or ping-pong algorithms, which will significantly reduce the spurious DC term and provide uniform power in the two designed focal spots (Figure 9b). The undesired additional focal spots are also formed and are associated with the fact that there are two off-axis points in the designed plane (no symmetry across the optical axis). Thus, even after iterative optimization, their influence cannot be neglected.

A totally different solution can be achieved by using the phase structure obtained by diffractive neural network, which makes the strong and synergistic connection between deep learning methods and the propagation based on the angular spectrum of plane waves formulation. In Figure 9c a single-plane phase distribution of the structure obtained by neural network design is presented in grey scale (bottom panel). The phase distribution does not resemble the shape expected by an optical intuition. Therefore, two shifted and combined lens-like structures will be expected as illustrated in Figure 9a. On the contrary, the phase distribution given by neural network design has a totally different shape, which is not predictable using classic diffractive design methods. The tremendous difference between different design approaches can be observed in the reconstructed intensity pattern where the two focal spots have almost identical shape as in the iterative algorithm, but spurious spots are not visible. This gives the enormous superiority of neural-network-based design, especially for complex desired intensity patterns (designed in off-axis approach and being not symmetrical to the optical axis of the element).

## 9. Conclusions

Terahertz radiation is at the boundary of two different domains—optics and electronics. This is still not a fully discovered area that is entwining physical phenomena describing two permeating worlds and interfering laws of optics and electronics to constitute the uniform entirety of this unique and beautiful THz world. The fusion of two superposing physical worlds becomes an enormous challenge in designing optical elements. Moreover, we need to fight against the tremendous diffraction effects resulting from small aperture setups in comparison with the wavelength of propagating radiation. At the same time, we need to battle with ultracoherent radiation sources entailing dreadfully disturbing interference and at the same time struggling with a merciless broad range of THz frequencies. This specific and thrilling environment makes the THz optics magic as it unifies two miscellaneous fields of the beautiful world of physics.

Profiting from a variety of terahertz optical materials, the simplicity and accessibility of manufacturing techniques and relatively long wavelength size—terahertz diffractive optical elements give immense possibilities of application. The design method can be readily scaled up to different frequency ranges (also to optical or to microwave) and even within the terahertz range the two poles can be differentiated: the first, implementing 3D printing technology and different polymer-like materials and the second one exploiting more advanced manufacturing techniques based on silicon (like etching or ablation, remembering about anti-reflective structures). These two groups form two different terahertz worlds, which can jointly permeate each other by using transparent materials (being dispersion free in the terahertz range) together with high order kinoform design. It is a very attractive solution that could expand cost-efficient manufacturing of good quality optical elements at higher terahertz frequencies.

The enormous dominance of diffractive optics lies in the ability to realize large apertures, short focal lengths and thin elements, especially in THz range. Such elements can allow going ahead with imaging or scanning system applications where a plane to plane imaging must be performed by the optical system. The raster scanning methods will never defeat the speed of instantaneous forming of the image in the whole plane. Moreover, the diffractive design also favors off-axis functioning or developing more complicated optical structures or focal spots as well as redirecting the incident radiation. The whole diversity of optical solutions given by diffractive optics can be enriched and improved by using neural network design leading to deep learning or artificial intelligence based optical systems.

## Figures and Tables

**Figure 1 sensors-21-00100-f001:**
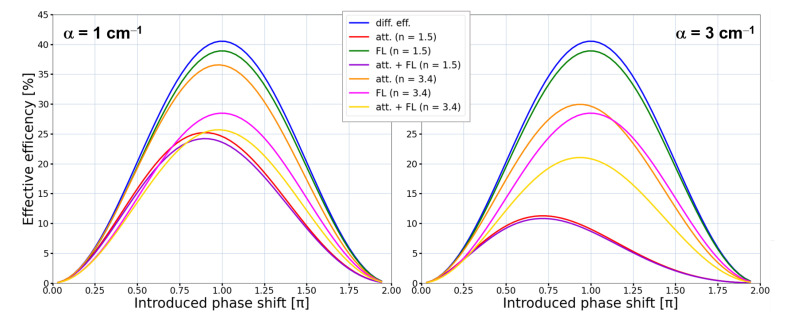
The effective efficiency of the binary phase diffractive element for two different absorption coefficient values: α = 1 cm${}^{-1}$ (**left** diagram) and α = 3 cm${}^{-1}$ (**right** diagram). The effective efficiency values are given under assumption of considering only diffractive efficiency (diff. eff.), attenuation introduced by the thickness of the structure resulting from absorption coefficient value (att.), Fresnel losses resulting from the reflection of part of the radiation of the interface between two media with different refractive indices (FL) and considering both—attenuation coefficient influence and Fresnel losses (att. + FL). It can be noticed that for materials with larger attenuation the effective efficiency is larger for structures with smaller phase shift than resulting from theoretical diffraction efficiency due to the attenuation of material. The influence of Fresnel losses is increasing with the increase of the refractive index value.

**Figure 2 sensors-21-00100-f002:**
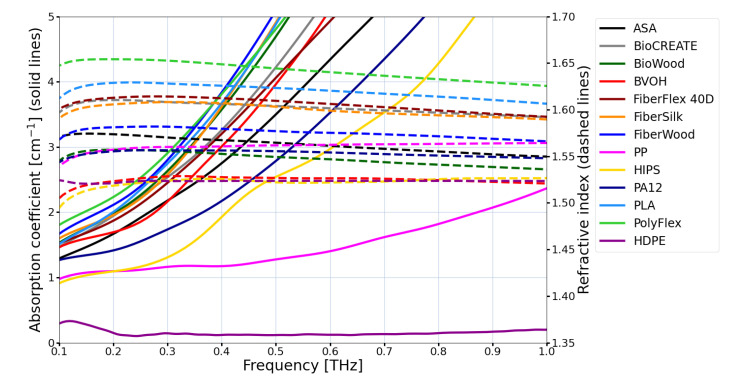
Refractive indices and absorption coefficients of exemplary materials available for extrusion-based 3D printing technique. High density polyethylene (HDPE) is illustrated as a reference material but it cannot be processed using this technique. The group of biopolymers—FiberWood, FiberSilk, BioCREATE and BioWOOD, thermoplastic polyurethane (PolyFlex) and thermoplastic polyester elastomer (FiberFlex 40D) are characterized with largest absorption coefficient values—similar to polylactide (PLA). The butenediol vinyl alcohol co-polymer (BVOH) together with acrylonitrile styrene acrylate (ASA) have slightly smaller absorption coefficient value. The most transparent (not taking into account reference HDPE) is polypropylene (PP), then high impact polystyrene (HIPS) and nylon–poliamid 12 (PA12). The refractive index values for all illustrated materials are in the range between 1.5 and 1.65, which is very similar to the values of refractive index for glass for visible light.

**Figure 3 sensors-21-00100-f003:**
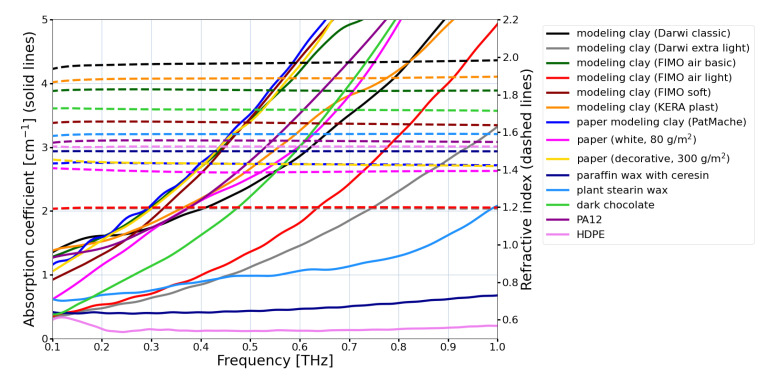
Refractive indices and absorption coefficients of alternative THz optical materials. Different types of modelling clay with refractive index values ranging from 1.2 up to 2, papers, paraffin, chocolate. HDPE and PA12 are plotted as reference materials.

**Figure 4 sensors-21-00100-f004:**
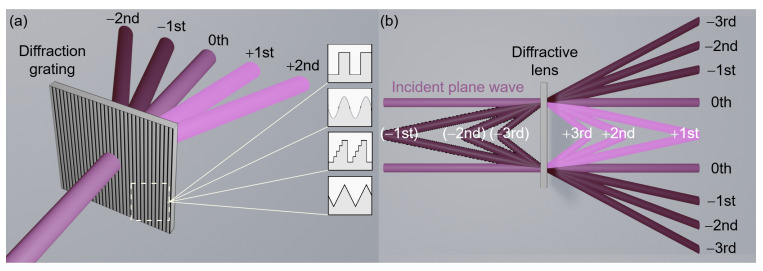
(**a**) The scheme of forming diffraction orders by the grating together with different diffraction grating types—different coding types: binary, sinusoidal, step-like, kinoform. (**b**) The scheme of focusing efficiency of diffractive converging lens. Colors corresponds to particular diffraction grating orders.

**Figure 5 sensors-21-00100-f005:**
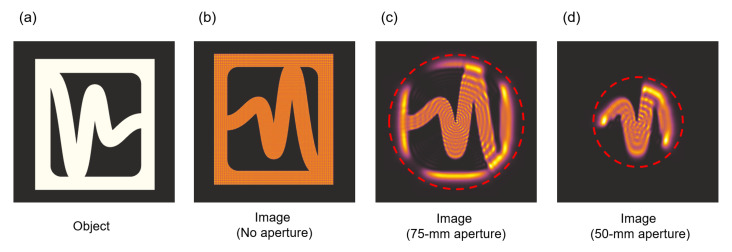
(**a**) The exemplary object used in simulations for 0.3 THz. (**b**) The image obtained in 4f imaging setup assuming no aperture, only calculation matrix size. (**c**) The image obtained in 4f imaging setup assuming lens apertures limited to 75 mm (marked with red dashed line) (**d**) The image obtained in 4f imaging setup assuming lens apertures limited to 50 mm (marked with red dashed line). Formed images are inverted as in 4f setups. The presented area corresponds to 90-mm-square area cut from 204.8-mm calculation matrix.

**Figure 6 sensors-21-00100-f006:**
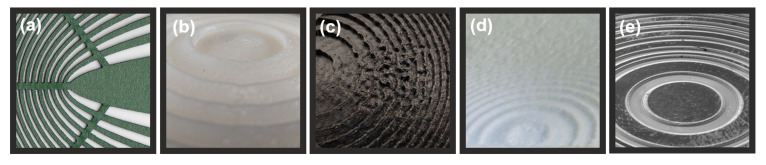
Different THz diffractive optical structures manufactured using: (**a**) laser cutting to manufacture hyperbolic lens [9] from paper, (**b**) extrusion-based 3D printing to make a mold to form converging lens from paraffin, (**c**) extrusion-based 3D printing from PA12 for off-axis focusing lens, (**d**) selective laser sintering (powder-based 3D printing) from PA12 to manufacture point-to-point redirecting lens and (**e**) ablation in silicone to fabricate Fibonacci lens (photograph courtesy Linas Minkevičius, FTMC, Vilnius).

**Figure 7 sensors-21-00100-f007:**
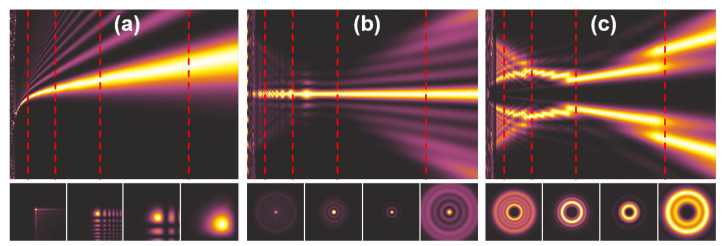
The intensity patterns of: (**a**) Airy, (**b**) Bessel and (**c**) optical vortex beams. The upper panel illustrates the propagation along optical axis—yz plane, while the lower panel corresponds to the xy cross-sections at distances after optical element marked with red dashed lines. The structure is located on the left edge of yz distributions.

**Figure 8 sensors-21-00100-f008:**
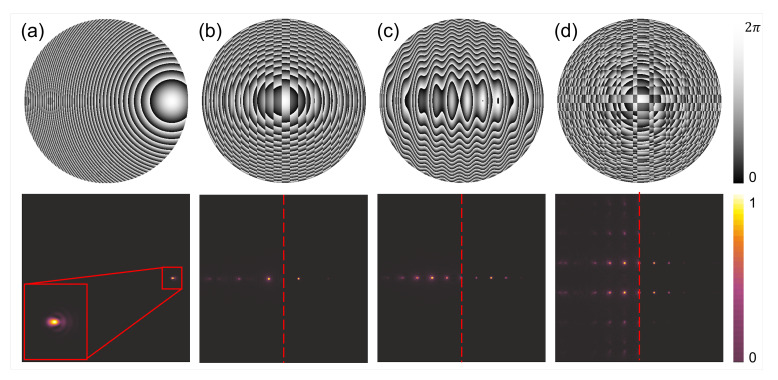
Different diffractive structures redirecting and splitting the incident radiation. The upper panel illustrates the phase delay maps introduced by each structure. The lower panel illustrates the intensity distribution in the focal plane in (**a**). Additionally, due to the fact the distributions are symmetrical in (**b**–**d**) the focal plane is illustrated as two halves—left corresponding to amplitude distribution and right the intensity distribution. The structure in (**a**) is a shifted lens to redirect and focus the radiation out of the optical axis. The inset shows the magnification of focal spot which will have more aberrations when shifted further from the optical axis. The structure in (**b**) is the binary phase grating joined with the converging lens. It forms two strong spots aside optical axis. The structure in (**c**) is the sinusoidal phase grating joined with the converging lens. Due to the modified changes of the phase the spots corresponding to other order of diffraction than ±1st are also visible. The structure in (**d**) is the Dammann grating forming the matrix of 3 × 2 points joined with the converging lens. In this case additional orders are very noticeable. Thus, in case of redirecting the beam there in a need to optimized the phase delay map to suppress the influence of undesired spurious focal spots.

**Figure 9 sensors-21-00100-f009:**
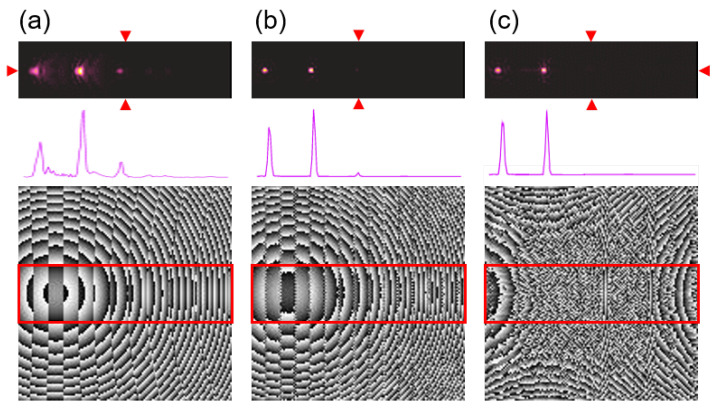
The comparison of different DOE design methods—(**a**) theoretical off-axis lens equations with 25 mm and 50 mm shifts, (**b**) iterative algorithm and (**c**) neural network optimization of phase distribution. The calculated phase distributions of the diffractive structures are shown as a grey scale image (lower panel) corresponding to introducing phase retardation from 0 (black) to 2π (white). On the upper panel the corresponding reconstructed intensity patterns are given in color palette. The reconstructed area was the same size as the structure, but here was limited to the region marked with red rectangles on phase retardation maps. The cross-sections along the horizontal axis crossing the main optical axis (marked with red triangles) are shown in the middle panel.

**Table 1 sensors-21-00100-t001:** The maximal values of diffraction efficiency of structures $\eta_{1}$ corresponding to different coding types and the numerical equations describing the efficiency in different orders of diffraction. *—$J_{1}$ is the Bessel function of first kind and first order. **—Δh is the contrast (amplitude) of the grating changes.

Phase Coding Method	$\eta_{1}$	$\eta_{m}$
Kinoform (1st order)	100%	$sinc^{2}\left(1-m\right)$
Kinoform ($p^{th}$ order)	100%	$sinc^{2}\left(p-m\right)$
Phase N-level	up to 100%	$sinc^{2}\left(\frac{m}{N}\right)$
Phase 16-level	99%	$sinc^{2}\left(\frac{m}{16}\right)$
Phase 8-level	95%	$sinc^{2}\left(\frac{m}{8}\right)$
Phase 4-level	81%	$sinc^{2}\left(\frac{m}{4}\right)$
Phase binary (2-level—with fill factor *a*)	up to 40.4%	$\frac{4}{{\left(\pi m\right)}^{2}}sin^{2}\left(\frac{\phi }{2}\right)sin^{2}\left(\pi ma\right)$
Phase sinusoidal *	up to 33.8%	$J_{1}^{2}\left(\frac{\phi }{2}\right)$
Amplitude binary (with fill factor *a*)	up to 10.1%	$\frac{sin^{2}\left(\frac{\pi am}{2}\right)}{{\left(\pi m\right)}^{2}}$
Amplitude sinusoidal **	up to 6.3%	${\left(\frac{\Delta h}{2}\right)}^{2}$

## Data Availability

The data presented in this study are available on request from the corresponding author. The data are not publicly available due to the qualitative nature and despite all transcribed data are being kept in a different form.

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
