# Peer review of "The Magic of Optics—An Overview of Recent Advanced Terahertz Diffractive Optical Elements"

_sensors, 2020, doi:10.3390/s21010100_

Round 1

Reviewer 1 Report

The author presents a review of diffractive optics in the terahertz range. The review is thorough, well-structured, and it covers a topic that is of great importance to the development of terahertz technology. 

There are several issues that must be addressed for this manuscript to merit publication as a review article in Sensors

  1. The opening of the title, namely the phrase “The magic of optics,” should be omitted. It seems contrived, vague, nonspecific, and overly poetic and prosaic for the subject matter of optics. 
  2. The review is focused on diffractive optics for which the required phase distribution is obtained by control of path-length, i.e. by some particular topology of refractive or reflective material. However, there is another category of terahertz diffractive optic that the author has entirely overlooked. Spatially-dependent phase-change can be acquired instantaneously, by interaction with a resonator, in order to achieve beam-steering of terahertz radiation. Devices that employ this technique are variously called reflectarrays, transmitarrays, and nonuniform metasurfaces, and have received significant attention in the past decade. I am not suggesting to change the focus of the author’s review, but at present, the lack of information about resonator-based terahertz optics seems a somewhat glaring omission. The author should make some mention of resonator-based diffractive optics, and compare and contrast them with the topological, path-length approach that is favoured in the review. Relevant references are (Science 340(6138), 1304-1307, 2013; Appl. Phys. Lett. 103, 041109, 2013; Proc. SPIE 8715, 871506, 2013; Opt. Commun. 322, 164–168, 2014; ACS Photonics 3(6), 1019–1026, 2016; Opt. Lett. 42(9), 1867-1870, 2017; Opt. Express 26(11), 14132-14142, 2018). 
  3. In the Introduction, the author should justify why optics are critical for terahertz waves in general. For example, is it because high-gain aperture antennas are required to mitigate path loss? Is it because of a lack of waveguiding technologies, thereby necessitating the guiding manipulation of terahertz waves in free-space?
  4. In Sections 1 and 2, there is inadequate citation, and this leads to several statements that are unsubstantiated. For instance, on line 18, the author asserts that diffractive optics “can look like Fresnel lenses, spirals, map[s] of mountains, peacock eye[s]...”, or on line 41, that “DOEs can be used as contact lenses.” The author should provide specific references for each such assertion. At present, the first citation is on line 54, on the second page, and pertains to the use of anti-reflection coatings. This is an unusual choice of starting point to introduce academic literature that is relevant to the topic of the review. 
  5. Several citations (7,26 44, 47, 54, 63, 66, 67, 74, 83) are for conference papers. It is always preferable to cite a journal article wherever possible, as they are more detailed and complete resources on a given topic. This is of key interest to readers of review articles, as articles of this sort can be utilized as an entry point into comprehension of the academic literature of a field as a whole. For this reason, the citations of conference papers should be replaced with journal articles, wherever possible.
  6. Figure 2 provides the properties of several materials that may be shaped by 3D printing, either directly, or by casting in a 3D-printed form. As such, 3D-printing is implicitly highlighted as a fabrication technique that is of key interest to this field—no other technique receives this treatment in the manuscript. For this reason, the author may wish to note that direct 3D printing of all-metal terahertz components has also been demonstrated in the literature (Opt. Express 24(15), 17384-17396, 2016; J. Infrared, Millim. Terahertz Waves 39, 535–545, 2018)).
  7. The wording of several of the descriptions and explanations could be improved. For instance, the introduction opens with “Diffractive optical elements (DOEs) are structures with transmittance defined by complex function that diffracts the radiation to obtain a particular phase distribution after it.” The use of the term “transmittance defined by complex function” gives the distinct impression of a two-port network, rather than a spatially-dependent phase shift, and so the means whereby a “distribution” is produced may be unclear to readers. Furthermore, the complex function does not “diffract the radiation to obtain a phase distribution,” but rather, the converse is true. A phase distribution is imposed in order to diffract an outgoing beam in a particular way. As this is a review paper, it is intended to be a reliable reference on this topic, and hence it is of critical importance that all such explanations are as clear and accurate as possible. 
  8. The manuscript could be improved with the inclusion of a greater number of figures. At present, there are seven figures in total, which is somewhat low for a review paper. More figures could break up the text, and make it more engaging and readable. 
  9. At times, the English language is informal, and not up to standard for academic publication. I recommend to enlist the services of a native speaker of English for proofreading.

Author Response

__________________________________________________________________________
Reviewer 1

Dear Reviewer,

The author would like to thank the reviewer for his/her positive in-depth feedback regarding my submitted paper. Below I address point by point the referee remarks and show how they were taken into account in the modified version of the paper.

Sincerely,
Agnieszka Siemion

____
The author presents a review of diffractive optics in the terahertz range. The review is thorough, well-structured, and it covers a topic that is of great importance to the development of terahertz technology.

Reply: Thank you very much.

____
There are several issues that must be addressed for this manuscript to merit publication as a review article in Sensors.
The opening of the title, namely the phrase “The magic of optics,” should be omitted. It seems contrived, vague, nonspecific, and overly poetic and prosaic for the subject matter of optics.

Reply:
Maybe this part of the title is not scientific in all meanings, but the Author wanted to make the title and the article not boring and fitted to schedules. The Author is strongly convinced to keep also some lightness and easiness in describing physical/optical phenomena thus the title remained without changes. Such decision was taken by consultation within Authors research group and taking into account also the remarks of the other Reviewer.

____
The review is focused on diffractive optics for which the required phase distribution is obtained by control of path-length, i.e. by some particular topology of refractive or reflective material. However, there is another category of terahertz diffractive optic that the author has entirely overlooked. Spatially-dependent phase-change can be acquired instantaneously, by interaction with a resonator, in order to achieve beam-steering of terahertz radiation. Devices that employ this technique are variously called reflectarrays, transmitarrays, and nonuniform metasurfaces, and have received significant attention in the past decade. I am not suggesting to change the focus of the author’s review, but at present, the lack of information about resonator-based terahertz optics seems a somewhat glaring omission. The author should make some mention of resonator-based diffractive optics, and compare and contrast them with the topological, path-length approach that is favoured in the review. Relevant references are (Science 340(6138), 1304-1307, 2013; Appl. Phys. Lett. 103, 041109, 2013; Proc. SPIE 8715, 871506, 2013; Opt. Commun. 322, 164–168, 2014; ACS Photonics 3(6), 1019–1026, 2016; Opt. Lett. 42(9), 1867-1870, 2017; Opt. Express 26(11), 14132-14142, 2018).

Reply:
Yes, the review is focused on diffractive optics as it is its scope. It was supposed to describe the recent advanced optical elements in the form of tutorial. Thus, the resonator based optical elements were not introduced here not to complicate the topic.
However, according to the Reviewer advise an additional sentences were added in the introduction with suggested reference as well as additional kind of tutorial/review works:

The another possibility to precisely control the phase distribution is related to a little bit different type of optical elements. This group is based on the interaction of THz radiation with array of resonators which form spatially dependent phase changes resulting in required beam steering. Such elements are described as reflect-arrays, transmit-arrays, nonuniform meta-surfaces or meta-materials. In the last years the resonator-based optics is gaining huge interest of researchers [13–18]. Although, such devices also introduce a particular phase distribution corresponding to phase delay map they function according to different phenomena than described in this work diffractive elements. More information about different methods of terahertz beam forming was described in [19–21].

____
In the Introduction, the author should justify why optics are critical for terahertz waves in general. For example, is it because high-gain aperture antennas are required to mitigate path loss? Is it because of a lack of waveguiding technologies, thereby necessitating the guiding manipulation of terahertz waves in free-space?

Reply:
The aim of the Author was not to favor diffractive optical elements but to introduce them as an interesting and effective solution also for THz region. Author wanted to highlight differences and aspects of DOE designing for THz waves which is somehow different than for visible light (for example the aspect ratio between the size of elements and the wavelength are much different). Additional sentences were added in the introduction:

In many aspects, THz radiation range is peculiar due to the fact that it combines two entangled worlds - of optics and electronics. This is the region where solutions utilized in both these domains slightly stop to be effective. Antenna based systems, multiplexers, waveguides, couplers, phased arrays, Luneburg lenses and other optical devices used for millimetre waves are meeting with the world of mirrors, refractive and diffractive lenses, fibres, prisms, gratings to form a unique mixture of optical and electronic phenomena. For different applications depending on design frequency (lower or higher part of THz region) various optical solutions can be used. Similar to the use of emitters/sources and detectors having their roots in photonics or electronics, the optical devices are also transformed to THz region and are not favoring any design technique. Depending on the assumed utilization and the design frequency range different solutions should be used, either approaching optical devices or taking advantage of electronic components.

____
In Sections 1 and 2, there is inadequate citation, and this leads to several statements that are unsubstantiated. For instance, on line 18, the author asserts that diffractive optics “can look like Fresnel lenses, spirals, map[s] of mountains, peacock eye[s]...”, or on line 41, that “DOEs can be used as contact lenses.” The author should provide specific references for each such assertion. At present, the first citation is on line 54, on the second page, and pertains to the use of anti-reflection coatings. This is an unusual choice of starting point to introduce academic literature that is relevant to the topic of the review.

Reply:
References that were used later were also introduced in the beginning of the paper. Moreover, the additional references were also added. (First 29 references in the revised article.)

____
Several citations (7,26 44, 47, 54, 63, 66, 67, 74, 83) are for conference papers. It is always preferable to cite a journal article wherever possible, as they are more detailed and complete resources on a given topic. This is of key interest to readers of review articles, as articles of this sort can be utilized as an entry point into comprehension of the academic literature of a field as a whole. For this reason, the citations of conference papers should be replaced with journal articles, wherever possible.

Reply:
The Author tried to replace conference papers, however, not everywhere it was possible.
Citation 7 was replaced with:
Squires, A. D., & Lewis, R. A. (2018). Feasibility and characterization of common and exotic filaments for use in 3D printed terahertz devices. Journal of Infrared, Millimeter, and Terahertz Waves, 39(7), 614-635.

Citation 44 was replaced with:
Minkevičius, L., Indrišiūnas, S., Šniaukas, R., Račiukaitis, G., Janonis, V., Tamošiūnas, V., ... & Valušis, G. (2018). Compact diffractive optics for THz imaging. Lithuanian Journal of Physics, 58(1).

Additional citations were added to reference 47:
Zhao, F., Li, Z., Dai, X., Liao, X., Li, S., Cao, J., ... & Li, H. (2020). Broadband Achromatic Sub‐Diffraction Focusing by an Amplitude‐Modulated Terahertz Metalens. Advanced Optical Materials, 8(21), 2000842.
Yang, M., Ruan, D., Du, L., Qin, C., Li, Z., Lin, C., ... & Wen, Z. (2020). Subdiffraction focusing of total electric fields of terahertz wave. Optics Communications, 458, 124764.

Citation 54 was replaced with:
Liebert, K., Rachoń, M., Bomba, J., Sobczyk, A., Zagrajek, P., Sypek, M., ... & Siemion, A. (2018). THz diffractive focusing structures for broadband application. Photonics Letters of Poland, 10(3), 76-78.
Liebert, K., Rachon, M., Kolodziejczyk, A., Sypek, M., Ducin, I., ZAGRAJEK, P., & Siemion, A. (2020). Study of thin, achromatic diffractive structures to focus terahertz radiation on a detector. Optica Applicata, 50(3).

Citation 66 was replaced with:
Wu, G. B., Chan, K. F., & Chan, C. H. (2020). 3-D Printed Terahertz Lens to Generate Higher-Order Bessel Beams Carrying OAM. IEEE Transactions on Antennas and Propagation.

Citation 67 was replaced with:
Choporova, Y. Y., Knyazev, B. A., Kulipanov, G. N., Pavelyev, V. S., Scheglov, M. A., Vinokurov, N. A., ... & Zhabin, V. N. (2017). High-power Bessel beams with orbital angular momentum in the terahertz range. Physical Review A, 96(2), 023846.

Additional citation was added to reference 83:
Sourangsu, B., & Berardi, S. R. (2019). A Computational Design Framework for Efficient, Fabrication Error-Tolerant, Planar THz Diffractive Optical Elements. Scientific Reports (Nature Publisher Group), 9(1).

____
Figure 2 provides the properties of several materials that may be shaped by 3D printing, either directly, or by casting in a 3D-printed form. As such, 3D-printing is implicitly highlighted as a fabrication technique that is of key interest to this field—no other technique receives this treatment in the manuscript. For this reason, the author may wish to note that direct 3D printing of all-metal terahertz components has also been demonstrated in the literature (Opt. Express 24(15), 17384-17396, 2016; J. Infrared, Millim. Terahertz Waves 39, 535–545, 2018)).

Reply:
The additional sentences were added to the article text:
Many different 3D printed materials were analyzed due to the simplicity and accessibility of manufacturing DOEs for THz range. Moreover, the direct 3D printing of all-metal terahertz components has also been demonstrated in the literature [55, 56].
____
The wording of several of the descriptions and explanations could be improved. For instance, the introduction opens with “Diffractive optical elements (DOEs) are structures with transmittance defined by complex function that diffracts the radiation to obtain a particular phase distribution after it.” The use of the term “transmittance defined by complex function” gives the distinct impression of a two-port network, rather than a spatially-dependent phase shift, and so the means whereby a “distribution” is produced may be unclear to readers. Furthermore, the complex function does not “diffract the radiation to obtain a phase distribution,” but rather, the converse is true. A phase distribution is imposed in order to diffract an outgoing beam in a particular way. As this is a review paper, it is intended to be a reliable reference on this topic, and hence it is of critical importance that all such explanations are as clear and accurate as possible.

Reply:
The description was changed and now it reads:
Diffractive optical elements (DOEs) are structures that introduce particular phase shifts of the incoming wave to create a desired phase profile of the outgoing beam. Such structures are defined by their transmittance which is defined by a complex function. This function is transformed into an optical element that diffracts the radiation to obtain a particular phase distribution after it. The complex transmittance describes how the amplitude and the phase of the incoming radiation will be changed after passing through the structure.

____
The manuscript could be improved with the inclusion of a greater number of figures. At present, there are seven figures in total, which is somewhat low for a review paper. More figures could break up the text, and make it more engaging and readable.

Reply:
The Author added two figures that could break the text and help in understanding.
Current Fig. 1 and Fig. 8 were added with descriptive captions.

____
At times, the English language is informal, and not up to standard for academic publication. I recommend to enlist the services of a native speaker of English for proofreading.

Reply:
English grammar has been verified. Text was already corrected by Foreign Language Division at Warsaw University of Technology by the English academic teacher who is bilingual.
The additional check by the native speaker may be conducted on demand of the reviewer.

Reviewer 2 Report

In this article the author reviews the sate of the art for terahertz diffractive optical elements. The article is technically correct and it can give the reader a good background for this particular subject. I recommend the article for publication after minor revision.

Here I give some comments, suggestions:

  1. The introduction is too short. Since the article is an overview of recent advances, a presentation of some of them should be exposed in the introduction.
  2. There is a lack of references in the first three paragraphs of section 2.
  3. The second paragraph of section 2 is somewhat confusing. Please rewrite it.
  4. Figure caption of Figure 3. I believe that much part of the text fits more properly in the main text of the article than in the caption.
  5. Line 101: “0.05mm” In the use of units, it must be taken into account that the units should not be expressed in italics. The correct expression is : “0.05 mm”
  6. I miss a reference in the first paragraph of section 3.
  7. Equation (4): f(x) and d are not defined. I assume f(x) is the “function describing the periodic grating” (paragraph before equation (3)), and d is the period of the grating. But these ones must be explained in the text.
  8. The equation linking the extinction coefficient and the attenuation coefficient should be put in a different paragraph (line).
  9. Line 179: “λ = 0.5µm” must be “λ = 0.5 µm. As in comment 5. Please check, since this mistake is repeated throughout the text.

10 In equation 8 appears θ, but in the text it is referred as θ, please check.

11 Φ of equation (11) is not explained.

Author Response

__________________________________________________________________________
Reviewer 2

Dear Reviewer,

The author would like to thank the reviewer for his/her positive in-depth feedback regarding my submitted paper. Below I address point by point the referee remarks and show how they were taken into account in the modified version of the paper.

Sincerely,
Agnieszka Siemion

____
In this article the author reviews the sate of the art for terahertz diffractive optical elements. The article is technically correct and it can give the reader a good background for this particular subject. I recommend the article for publication after minor revision.
Here I give some comments, suggestions:
The introduction is too short. Since the article is an overview of recent advances, a presentation of some of them should be exposed in the introduction.
There is a lack of references in the first three paragraphs of section 2.

Reply:
References that were used later were also introduced in the beginning of the paper. Moreover, the additional references were also added. (First 29 references in the revised article.)

____
The second paragraph of section 2 is somewhat confusing. Please rewrite it.

Reply: The paragraph has been rewritten to the following form: 'In case of the visible light the major drawback is that human eye can see all the lines corresponding to the phase discontinuities in the structure. The human eye is very sensitive for small changes of the refractive index. Therefore DOEs are used e.g. in the contact lenses (placed right after the pupil) but can hardly be used in the corrective glasses. In this spectral range, DOEs are also commonly used in the cameras and imaging setups. There is a whole variety of available optical materials, however manufacturing ensuring required resolution is rather expensive (taking into account for example electron beam lithography, laser lithography, etc.).'.

____
Figure caption of Figure 3. I believe that much part of the text fits more properly in the main text of the article than in the caption.

Reply: The caption of the Figure 3 has been shortened. Part of the removed content has been moved to the main text right before the figure.

____
Line 101: “0.05mm” In the use of units, it must be taken into account that the units should not be expressed in italics. The correct expression is : “0.05 mm”

Reply: Fixed here as well as in all other instances of this issue.

____
I miss a reference in the first paragraph of section 3.

Reply:
The following references were added in first two paragraphs of section 3:
O’Shea, D.C.; Suleski, T.J.; Kathman, A.D.; Prather, D.W.Diffractive optics: design, fabrication, and test; Vol. 62, SPIE press, 2004.
Goodman, J.W.Introduction to Fourier optics; Roberts and Company Publishers, 2005
del Mar Sánchez-López, M.; Moreno, I.; Martínez-García, A. Teaching diffraction gratings by means of a phasor analysis. Education and Training in Optics and Photonics. Optical Society of America, 2009, p.EMA1.
Jiménez, J.; Anera, R.; Jiménez del Barco, L.; Hita, E. Effect on laser-ablation algorithms of reflection losses577and nonnormal incidence on the anterior cornea. Applied Physics Letters 2002, 81, 1521–1523.
Soifer, V. A., Kotlar, V., & Doskolovich, L. (1997). Iteractive Methods For Diffractive Optical Elements Computation. CRC Press.
Wyrowski, F. (1990). Diffractive optical elements: iterative calculation of quantized, blazed phase structures. JOSA A, 7(6), 961-969.

____
Equation (4): f(x) and d are not defined. I assume f(x) is the “function describing the periodic grating” (paragraph before equation (3)), and d is the period of the grating. But these ones must be explained in the text.

Reply: Description in the text has been reformulated to: 'Gm is the amplitude coefficient of the expansion of the function f(x) describing the periodic grating with the period d into the Fourier series given as:'

____
The equation linking the extinction coefficient and the attenuation coefficient should be put in a different paragraph (line).

Reply: The equation has been placed in the separate line.

____
Line 179: “λ = 0.5µm” must be “λ = 0.5 µm”. As in comment 5. Please check, since this mistake is repeated throughout the text.

Reply: The issue has been fixed here as well as in all other places.

____
10 In equation 8 appears θ’, but in the text it is referred as θ, please check.

Reply: The symbol at the end is a coma ',' separating the equation from the next line.

____
11 Φ of equation (11) is not explained.

Reply: Φ has been defined after the equation: 'Φ corresponds to the azimuthal angle in the structure's plane'

Round 2

Reviewer 1 Report

Remove "The magic of optics."

Provide practical justification for the necessity of optics for applications of terahertz waves.

Another issue is that the author has recently published a review paper on the same topic (https://doi.org/10.1007/s10762-019-00581-5). Is the manuscript under review not essentially redundant therewith? Why is it necessary for a single author to publish two review articles approximately one year apart? It is highly doubtful that there have been sufficiently many new developments in the field in the intervening time to warrant another review. If the author simply wishes to provide a new perspective on the topic then this is insufficient grounds to merit publication as a review article.

Author Response

Dear Reviewer,
Please find below my answers to the your remarks.
I am very grateful for useful insights and suggestions. 

_________________
Reviewer 1: Remove "The magic of optics."
Answer:
In principle, I could agree to remove the first part of the title. My motivation was the following: just to indicate that diffractive optics can be beautiful and useful tool in designing compact THz imaging system. But if you insist, I will remove.
However, such word is used in similar connotation in titles in different journals like for example:
Nonlinear magic: multiphoton microscopy in the biosciences - Nature Biotechnology
The magic of cluster SIMS - ACS Publications
Magic features of 68Ni - Physics Letters B
The Magic of ELFs - Journal of Cryptology (Springer)
The Magic of Duplicates and Aggregates (1990) – VLDB (Springer)
The magic of the sugar code - Trends in Biochemical Sciences (Elsevier)
The magic of the hypoxia-signaling cascade - Cellular and Molecular Life Sciences (Springer)
Magic identities for conformal four-point integrals - IOP Publishing Ltd
And even as a book title:
High-speed Signal Propagation: Advanced Black Magic - Prentice Hall Professional.

_________________
Reviewer 1: Provide practical justification for the necessity of optics for applications of terahertz waves.
Answer:
Practical justification is the following – further evolution of imaging and spectroscopy systems exhibits a trend of compact solutions because of convenience in use and reliability. As a rule, compact room temperature emitters and detectors are of the main focus of the scientific attention, however, passive elements like lenses, mirrors, splitters received much less emphasis. What is important in this context, optical components are relatively bulky (for instance, parabolic mirrors or beam splitters). Thus, diffraction optics can be one of the rational solutions in reducing the size of the system, on the other hand, such flat optical components can be integrated with THz sensors on a single chip making system free of precise optical alignment in its recording part. Moreover, for THz, diffractive optical elements are light and thin and they enable to realize large apertures and small focal lengths, which is of particular importance for example in SOFIA flying observatory missions (guided by European Space Agency ESA) for space observation. They also can do strongly off-axis beam shaping. And form different shapes and redirect almost freely. These features are quite attractive for further developments in THz imaging and spectroscopy.

Additional text was added to the manuscript text:
It should be underlined that diffraction optical elements are lightweight and thin. Additionally, they enable realization of the large aperture and small focal length elements. Moreover, they also are capable of strongly off-axis beam shaping and can form different shapes and can almost freely redirect the incident radiation. These features are quite attractive for further developments in THz imaging and spectroscopy.

_________________
Reviewer 1: Another issue is that the author has recently published a review paper on the same topic (https://doi.org/10.1007/s10762-019-00581-5). Is the manuscript under review not essentially redundant therewith? Why is it necessary for a single author to publish two review articles approximately one year apart? It is highly doubtful that there have been sufficiently many new developments in the field in the intervening time to warrant another review. If the author simply wishes to provide a new perspective on the topic then this is insufficient grounds to merit publication as a review article.
Answer:
It so happened that I was asked to write the review. Not a conventional review, but a review oriented mainly for sensors scientific and technological community within a Special Issue "Terahertz Imaging and Sensors". The idea was to demonstrate for the community possibilities of flat optics in future trends of THz imaging in reduction of systems size and reveal optical routes in making systems solid-state based or even realise on-chip solutions. Such a way means absence of bulky optics and optical alignment, however, it requires some other optics knowledge, in particular, if on-chip solution (optics plus sensors) are planned to be implemented. That is why many educational elements were introduced, that is why so many basics in design and characteristics were included – therefore, current review displays strong tutorial emphasis and highlights only very recent scientific results with exemplary materials for THz optics were not published before. Probably I will ask Editors to mark it as Tutorial Review – it would help for possible readers (scientists, technologists and circuits designers) to attract their attention to diffractive optics features and to use flat optics advantages in making sensors fully integrated with optical components providing thus new step in evolution in THz imaging systems.

The previous review was dedicated mainly to the THz community and concentrated on the state of the art of DOE for THz range starting from 1960's. It referred to 147 papers, out of which only 29 were published in the years 2016-2018. None of the references relates to works from 2019-2020.
In this review the recent research is described (2016-2020) - and there are the following articles that were not taken into account in previous review:

from 2020:
1. Rachon, M.; Liebert, K.; But, D.; Zagrajek, P.; Siemion, A.; Kolodziejczyk, A.; Sypek, M.; Suszek, J. Enhanced Sub-wavelength Focusing by Double-Sided Lens with Phase Correction in THz Range. JOURNAL OF INFRARED MILLIMETER AND TERAHERTZ WAVES 2020.
2. Siemion, A.; Komorowski, P.; Surma, M.; Ducin, I.; Sobotka, P.; Walczakowski, M.; Czerwi´nska, E. Terahertz diffractive structures for compact in-reflection inspection setup. Optics Express 2020, 28, 715–723.
3. Deuter, V.; Grochowicz, M.; Brose, S.; Biller, J.; Danylyuk, S.; Taubner, T.; Siemion, A.; Grützmacher, D.; Juschkin, L. Computational proximity lithography with extreme ultraviolet radiation. Optics express 2020, 28, 27000–27012.
4. Siemion, A.; Surma, M.; Komorowski, P.; Ducin, I.; Sobotka, P. Terahertz diffractive optics: different way of thinking. Terahertz Emitters, Receivers, and Applications XI. International Society for Optics and Photonics, 2020, Vol. 11499, p. 114990C
5. Castro-Camus, E.; Koch, M.; Hernandez-Serrano, A.I. Additive manufacture of photonic components for the terahertz band. Journal of Applied Physics 2020, 127, 210901.
6. Kononenko, T.V.; Knyazev, B.A.; Sovyk, D.N.; Pavelyev, V.S.; Komlenok, M.S.; Komandin, G.A.; Konov, V.I. Silicon kinoform cylindrical lens with low surface roughness for high-power terahertz radiation. Optics & Laser Technology 2020, 123, 105953
7. Zhao, F.; Li, Z.; Dai, X.; Liao, X.; Li, S.; Cao, J.; Shang, Z.; Zhang, Z.; Liang, G.; Chen, G.; others. Broadband Achromatic Sub-Diffraction Focusing by an Amplitude-Modulated Terahertz Metalens. Advanced Optical Materials 2020, p. 2000842.
8. Yang, M.; Ruan, D.; Du, L.; Qin, C.; Li, Z.; Lin, C.; Chen, G.; Wen, Z. Subdiffraction focusing of total electric fields of terahertz wave. Optics Communications 2020, 458, 124764
9. Liebert, K.; Rachon, M.; Kolodziejczyk, A.; Sypek, M.; Ducin, I.; ZAGRAJEK, P.; Siemion, A. Study of thin, achromatic diffractive structures to focus terahertz radiation on a detector. Optica Applicata 2020, 50.
10. Wu, G.B.; Chan, K.F.; Chan, C.H. 3-D Printed Terahertz Lens to Generate Higher-Order Bessel Beams Carrying OAM. IEEE Transactions on Antennas and Propagation 2020.
11. Yang, Y.; Ye, X.; Niu, L.; Wang, K.; Yang, Z.; Liu, J. Generating terahertz perfect optical vortex beams by diffractive elements. Optics Express 2020, 28, 1417–1425.
12. Seifert, J.M.; Hernandez-Cardoso, G.G.; Koch, M.; Castro-Camus, E. Terahertz beam steering using active diffraction grating fabricated by 3D printing. Optics Express 2020, 28, 21737–21744.
13. Liao, D.; Chan, K.F.; Chan, C.H.; Zhang, Q.; Wang, H. All-optical diffractive neural networked terahertz hologram. Optics Letters 2020, 45, 2906–2909

from 2019:
1. Hampel, B.; Tollkühn, M.; Elenskiy, I.; Martens, M.; Kajevic, D.; Schilling, M. Josephson Cantilevers for THz Microscopy of Additive Manufactured Diffractive Optical Components. IEEE Transactions on Applied Superconductivity 2019, 29, 1–4.
2. Surma, M.; Ducin, I.; Zagrajek, P.; Siemion, A. Sub-Terahertz Computer Generated Hologram with Two Image Planes. Applied Sciences 2019, 9, 659.
3. Siemion, A. Terahertz Diffractive Optics—Smart Control over Radiation. Journal of Infrared, Millimeter, and Terahertz Waves 2019, 40, 477–499. doi:10.1007/s10762-019-00581-5.
4. Knox, W.H. Inventing a New Way to See Clearly: Non-invasive Vision Correction with Femtosecond Lasers. Technology and Innovation 2019, 20, 385–398.
5. Fullager, D.B.; Park, S.; Hovis, C.; Li, Y.; Reese, J.; Sharma, E.; Lee, S.; Evans, C.; Boreman, G.D.; Hofmann, T. Metalized Poly-methacrylate Off-Axis Parabolic Mirrors for Terahertz Imaging Fabricated by Additive Manufacturing. Journal of Infrared, Millimeter, and Terahertz Waves 2019, 40, 269–275.
6. Indrišiunas, S.; Richter, H.; Grigelionis, I.; Janonis, V.; Minkevicius, L.; Valušis, G.; Raˇciukaitis, G.; Hagelschuer, T.; Hübers, H.W.; Kašalynas, I. Laser-processed diffractive lenses for the frequency range of 4.7 THz. Optics letters 2019, 44, 1210–1213.
7. Tofani, S.; Zografopoulos, D.C.; Missori, M.; Fastampa, R.; Beccherelli, R. Terahertz focusing properties of polymeric zone plates characterized by a modified knife-edge technique. JOSA B 2019, 36, D88–D96.
8. Shang, Y.; Wang, X.; Sun, W.; Han, P.; Ye, J.; Feng, S.; Zhang, Y. Terahertz image reconstruction based on compressed sensing and inverse Fresnel diffraction. Optics express 2019, 27, 14725–14735.
9. Iba, A.; Domier, C.W.; Ikeda, M.; Mase, A.; Pham, A.V.; Luhmann, N.C. Realizing sub-diffraction focusing for terahertz. 2019 44th International Conference on Infrared, Millimeter, and Terahertz Waves (IRMMW-THz). IEEE, 2019, pp. 1–2.
10. Pakhomov, A.; Arkhipov, R.; Arkhipov, M.; Demircan, A.; Morgner, U.; Rosanov, N.; Babushkin, I. Unusual terahertz waveforms from a resonant medium controlled by diffractive optical elements. Scientific reports 2019, 9, 1–12.
11. Zhang, Z.; Zhang, H.; Wang, K. Diffraction-free THz sheet and its application on THz imaging system. IEEE Transactions on Terahertz Science and Technology 2019, 9, 471–475.
12. Zhang, D.; Chen, B.; Ba, Z.; Ni, S.; Cao, J.; Wang, X. Generation of Broadband THz Airy Beams Applying 3D Printing Technique. 2019 13th European Conference on Antennas and Propagation (EuCAP). IEEE, 2019, pp. 1–3.
13. Minkevicius, L.; Jokubauskis, D.; Kašalynas, I.; Orlov, S.; Urbas, A.; Valušis, G. Bessel terahertz imaging with enhanced contrast realized by silicon multi-phase diffractive optics. Optics Express 2019, 27, 36358–36367.
14. Yang, Q.j.; Huang, J.g.; Xiao, Z.y.; Huang, Z.m.; Shu, R.; He, Z.p. Terahertz dispersion using multi-depth phase modulation grating. Optics express 2019, 27, 12732–12747.
15. Banerji, S.; Sensale-Rodriguez, B. 3D-printed diffractive terahertz optical elements through computational design. Micro-and Nanotechnology Sensors, Systems, and Applications XI. International Society for Optics and Photonics, 2019, Vol. 10982, p. 109822X.
16. Banerji, S.; Sensale-Rodriguez, B. A computational design framework for efficient, fabrication error-tolerant, planar THz diffractive optical elements. Scientific reports 2019, 9, 1–9.
17. Luo, Y.; Mengu, D.; Yardimci, N.T.; Rivenson, Y.; Veli, M.; Jarrahi, M.; Ozcan, A. Design of task-specific optical systems using broadband diffractive neural networks. Light: Science & Applications 2019, 8, 1–14

On of 2019 is mt review paper, so there are 16 new articles in the field on 2019 and 13 in 2020. Altogether it makes 29 new literature positions. It should be also noticed that 39 out of 118 references are relating to the state of the art (articles older than 2016) and 17 of them were added after suggestions of reviewers.

Moreover, in this review a novel concept of utilizing neural networks (with additional simulations after network training) is presented. I think iterative or neural-network-based design is novel for not symmetrical structures and it was addressed in this review.

Thanks again for your remarks, and I hope that these highlights induced more clarity in the article preparation context, presentation style and given emphasis.

Sincerely yours,
Agnieszka Siemion